# Relationship Between Physical Fitness Index and Body Mass Index: A Cross-Sectional Study in Serbian Students of Biomedical Sciences

**DOI:** 10.3390/jfmk10040449

**Published:** 2025-11-19

**Authors:** Aldina Ajdinović, Elvis Mahmutović, Emir Biševac, Zerina Salihagić, Teodora Safiye, Oliver Radenković, Ilma Čaprić, Raid Mekić, Slaviša Minić, Dejan Aleksić, Mina Lilić, Saša Bubanj

**Affiliations:** 1Department of Biomedical Sciences, State University of Novi Pazar, 36300 Novi Pazar, Serbia; aajdinovic@np.ac.rs (A.A.); ehmahmutovic@np.ac.rs (E.M.); ebisevac@np.ac.rs (E.B.); zsalihagic@np.ac.rs (Z.S.); oradenkovic@np.ac.rs (O.R.); icapric@np.ac.rs (I.Č.); rmekic@np.ac.rs (R.M.); minicvms@gmail.com (S.M.); 2Department of Psychology, State University of Novi Pazar, 36300 Novi Pazar, Serbia; teodoras@np.ac.rs; 3Department of Neurology, Faculty of Medical Sciences, University of Kragujevac, 34000 Kragujevac, Serbia; drdeal1987@gmail.com; 4Faculty of Science and Engineering, Sorbonne University, 75006 Paris, France; mina.lilic@etu.sorbonne-universite.fr; 5Faculty of Sport and Physical Education, University of Niš, 18000 Niš, Serbia

**Keywords:** body mass index, physical fitness, modified Harvard step test, students, Serbia

## Abstract

**Objectives:** Physical fitness is vital to sustaining the health of each individual and represents the level of readiness that allows them to perform everyday activities with sufficient energy. The aim of this research was to assess the physical fitness index and to determine its relationship with body composition. **Methods:** This research included 121 students of the State University of Novi Pazar, Serbia. The modified Harvard step test was used to assess physical fitness, and the body mass index was used to assess body composition. **Results:** Statistical analysis indicated that the physical fitness of students was not significantly satisfactory, given the large percentage of students with low-average and poor levels of physical fitness. A strong negative correlation between physical fitness index and body mass index was shown by Pearson (−0.720) and Spearman (−0.659) correlation coefficients with a *p*-value < 0.001. The results of the chi-square test (χ^2^(3) = 88.94, *p* < 0.001) also confirm this correlation. **Conclusions:** This study indicates widespread poor physical fitness among students and highlights the importance of regular exercise as a key factor for improving physical abilities. Given the relatively high prevalence of suboptimal prevalence of suboptimal physical fitness among university students, our findings represent a critical wake-up call for public health authorities, underscoring the urgent need for targeted interventions to reverse this trend and safeguard the health potential of the next generation.

## 1. Introduction

Physical fitness is vital to sustaining the health of every individual and represents the level of readiness that allows them to perform everyday activities with sufficient energy. Given that physical fitness develops and changes throughout life, it is influenced by many factors, including age, growth, and maturity [1]. Maintaining physical fitness is essential, as it enables daily functioning, contributes to overall health, both physical and mental, reduces the risk of disease, and improves quality of life [2,3].

As a key indicator of health, physical fitness includes various aspects such as strength, endurance, flexibility, and other components of physical fitness. The general condition of students is greatly influenced by their level of physical condition; therefore, maintaining physical condition is of essential importance for preserving health and well-being, as it improves the quality of life and facilitates the performance of daily tasks [4].

Overweight has become a major global health concern, with prevalence rising steadily in both developed and developing countries, affecting adults as well as children and adolescents. This requires an urgent need to change lifestyle habits by increasing physical activity levels and improving nutrition [5]. Lack of physical activity is common, especially among students [6]. Therefore, raising awareness of the importance of physical activity and a healthy lifestyle, as well as measuring the level of physical fitness in students, is of utmost importance because it helps them recognize and adopt healthy lifestyle habits in the early stages of their education, and this all emphasizes the need for further detailed examination of the relationship between body mass index (BMI) and physical fitness index (PFI) [7].

The motivation for conducting this study stems from the growing concern that unhealthy lifestyle habits, sedentary behavior, and excess body weight may directly impair physical fitness and functional capacity in young people [8]. Since adolescence is a sensitive developmental stage during which health-related behaviors are often formed and consolidated, understanding how BMI relates to physical fitness is crucial for creating effective preventive strategies and early interventions [9].

In our study, the modified Harvard Step Test (HST) was used to assess PFI as a reliable tool for assessing physical fitness and functional status [10]. It is a simple test that serves to assess the cardiovascular ability of an individual and estimates their level of fitness based on the pulse rate measured after performing the test [4]. The use of BMI is considered an important indicator of a person’s level of physical fitness by assessing their risk for many diseases [11].

The aim of this research was to evaluate the PFI of students and to examine its relationship with body composition, expressed through BMI.

It is expected that higher BMI values will be negatively associated with PFI, suggesting that overweight and obesity reduce the level of physical fitness among students.

Although numerous studies have investigated the relationship between BMI and physical fitness, few have examined this association in biomedical science students—a subgroup characterized by high academic workload but limited structured physical activity. This study, therefore, contributes context-specific evidence from Serbia, a region where comparable data are scarce, helping to inform university-level health promotion initiatives.

## 2. Materials and Methods

### 2.1. Participants

The conducted research included a sample of 121 students from the Department of Biomedical Sciences, State University of Novi Pazar, Serbia, of which 60 students were from the Rehabilitation study program and 61 students from the Sport and Physical Education study program.

Participants were recruited through class announcements and university email invitations. Out of 138 students initially screened, 121 met the inclusion criteria and agreed to participate (response rate 87.7%). Seventeen students were excluded due to incomplete data or professional sport involvement. Of the total number of participants, 67 (55.4%) were male, while 54 (44.6%) participants were female, indicating that both sexes are well represented in this study.

### 2.2. Inclusion/Exclusion Criteria

The inclusion criteria for participation in the study were: being a full-time student at the State University of Novi Pazar, aged 18–25 years, in good general health, and not suffering from any acute or chronic conditions that could limit physical activity.

The exclusion criteria were: professional involvement in sports, the presence of diagnosed cardiovascular, metabolic, or musculoskeletal disorders, as well as incomplete or invalid questionnaire/test data. Students who are professionally involved in sports were excluded from the research, which was important in order to avoid potential bias in the results due to the advanced physical condition and skills they possess. This decision also ensured the validity of the results, as BMI is not an appropriate measure in professional athletes because it does not take into account body composition (percentage of fat, muscle, and bone tissue), especially in sports where muscle mass is a key factor for success [12].

### 2.3. Study Design, Measures, and Procedures

A cross-sectional study design was adopted. The research was conducted from October 2024 to December 2024. The modified HST [13] was used to assess PFI [4], and BMI was used to assess body composition. As a cross-sectional study, our analysis captures associations rather than causal relationships; thus, the results should not be interpreted as evidence of directionality between BMI and physical fitness. All measurements were carried out in the morning hours to reduce the potential impact of circadian variations, and they were performed by the authors, who are experienced researchers trained in anthropometric and functional testing procedures. Prior to testing, participants were given instructions and a demonstration to ensure proper posture, cadence, pulse counting, and testing accuracy.

### 2.4. Modified HST Measurement

The Harvard Step Test was originally developed in a laboratory at Harvard University and requires a bench height of 20 inches (approximately 50 cm) for men and 18 inches (approximately 45 cm) for women. The step height used in this study was 45 cm for female participants and 50 cm for male participants, consistent with the classical HST standards. The stepping cadence was maintained at 30 steps per minute using a digital metronome (Soundbrenner Core, Berlin, Germany). The test was terminated if the participant was unable to maintain cadence for 15 s, experienced dizziness, shortness of breath, or heart rate exceeded 200 bpm. Physical Fitness Index (PFI) was categorized following the modified thresholds proposed by Ryhming and later adapted by Khurde et al. [4]: excellent (≥90), good (80–89), high average (65–79), low average (55–64), and poor (<55). During the test, the subject repeatedly ascends and descends the platform at a cadence of 30 steps per minute for a maximum duration of 5 min, or until fatigue prevents continuation. The total duration of the test, measured in seconds, is recorded. Immediately after exercise, the subject is seated, and heart rate is measured during the 1st, 2nd, and 3rd minutes of recovery. Heart rate was measured manually by trained assessors via radial pulse palpation for 30 s, multiplied by two to obtain beats per minute, during the 1st, 2nd, and 3rd minutes of recovery. All assessors completed a reliability training session before data collection, achieving an inter-rater correlation of 0.94 (pilot sample, n = 10). Previous research confirms high test–retest reliability of this modified protocol (ICC = 0.91) Burnstein et al., 2011 [14].

Based on these values, a fitness index is calculated using the following formula [15]:B = n × 100/2 × (P1 + P2 + P3),

B—number of points achieved; n—duration of the test in seconds; P—heart rate.

For body mass measurement, subjects stood barefoot on a body mass scale with their feet slightly apart. Height was measured using a Martins Anthropometer manufactured by GPM, Bellinzona, Switzerland (to the nearest 0.1 cm), while participants were standing upright, with heels together and feet slightly apart, from the top of the head to the tips of the toes. The measurements required maximum attention to ensure accuracy and precision.

The modified HST was selected due to its established validity for assessing cardiovascular endurance and recovery capacity in non-athletic young adult populations. It requires minimal equipment, can be administered in large groups, and correlates strongly with VO_2_max (r ≈ 0.80), making it a practical field tool for this setting [4].

### 2.5. Anthropometric Measurements

Based on the measurements obtained, BMI was calculated using a standard formula,BMI = Weight (kg)/Height (m^2^) 

Participants were divided into groups according to the World Health Organization classification [16]. Furthermore, students wore light clothing for the purposes of performing the HST. During the test, if any of the participants experienced breathing difficulties, fatigue, discomfort, or a heart rate above 200 beats per minute, the test was immediately terminated [17].

### 2.6. Statistical Analysis

The Excel program from the Microsoft Office suite was used for data entry, and statistical data processing was performed in SPSS, version 21.0 (SPSS Inc., Chicago, IL, USA). A priori power analysis (G*Power 3.1) indicated that a sample of 121 provides 80% power to detect correlations of |r| ≥ 0.25 at α = 0.05 (two-tailed). The observed effect (r = −0.720) exceeds this threshold, confirming adequate statistical power.

Measures of descriptive statistics included number (N), frequency, mean values, standard deviations, minimum, maximum, skewness, and kurtosis. The primary outcome was the correlation between continuous BMI and continuous PFI. To describe the relationship between the studied variables, Pearson’s and Spearman’s correlation coefficients were used, along with significance tests, as well as the chi-square test. Pearson’s correlation was applied due to normally distributed data, and Spearman’s to confirm robustness against potential deviations from normality.

The normality of the continuous variables BMI and PFI was examined using the Shapiro–Wilk and D’Agostino–Pearson tests, accompanied by visual inspection of histograms and Q–Q plots. Because both BMI and PFI deviated from perfect normality (Shapiro–Wilk *p* < 0.01), both parametric and nonparametric correlations were computed. The association between BMI and PFI was evaluated using Pearson’s product–moment correlation with 95% confidence intervals (CIs) derived from Fisher’s z-transformation and by nonparametric Spearman’s rank correlation with bootstrap 95% CIs (2000 resamples). Linearity was visually assessed by a scatterplot including a fitted linear regression line and a LOESS smoother. In addition, a multivariable linear regression model was fitted to adjust for potential confounding by sex and study program, with PFI as the dependent variable and BMI, sex, and program entered as predictors. Model diagnostics included checks for normality of residuals, multicollinearity (variance inflation factor, VIF < 2), and influential observations (Cook’s distance and studentized residuals). As a supplementary descriptive analysis, PFI categories were cross-tabulated against BMI groups defined by WHO criteria, and a chi-square test with Cramér’s V (with bootstrap 95% CI) was reported. Statistical significance was set at *p* < 0.05 (two-tailed), and all estimates are presented with 95% CIs.

### 2.7. Ethical Considerations

This study was conducted in accordance with the ethical standards of the committee responsible for human experimentation (institutional and national) and the Helsinki Declaration of 1975, as revised in 2013. Voluntary written and informed consent was obtained from each participant before enrollment in the study. The protocol of the study was approved by the Ethics Committee of the State University of Novi Pazar, Novi Pazar, Serbia (approval No. 1004/24; date of approval: 3 April 2024).

## 3. Results

The following table provides basic descriptive statistics for the research variables.

Statistical data indicate that there is a significant amount of variation in the HST scores, with most respondents falling into the middle range. On the other hand, data on students’ BMI revealed that the average BMI was within the normal range (18.5–24.9), with a moderate amount of variation, and none of the values were overly extreme (Table 1).

Both BMI and PFI distributions showed modest deviations from normality (BMI: W = 0.9625, *p* = 0.0019; PFI: W = 0.9484, *p* = 0.00015), particularly for PFI. Therefore, both parametric and nonparametric approaches were applied in the subsequent analyses (Figure 1).

Table 2 shows that out of a total of 121 students, 90 (74.4%) participants have a normal weight, and 31 (25.6%) are overweight. The BMI range in this sample (18.6–29.0 kg/m^2^) represents only normal-weight and overweight categories, limiting generalization to underweight or obese populations.

Out of a total of 121 respondents, 16 (13.2%) achieved a good result, 50 (41.3%) achieved a high average result, 23 (19.0%) achieved a low average outcome, and 32 (26.4%) had a poor PFI result.

As illustrated in Figure 2, the frequency distribution of PFI scores demonstrates the relative proportions of normal-weight and overweight students across different performance levels.

Figure 2 shows the BMI (kg/m^2^) among students at different levels of PFI, where the category of normal weight (18.5–24.9) had 16 subjects (13.22%) with a good (80–89) PFI result, 50 (41.32%) with a high average (65–79) close PFI, 20 (16.53%) with a low average (55–64) PFI, and 4 (3.31%) with a poor (<55) PFI score. Among the overweight (25–29.9), 3 (2.48%) respondents had a low average (55–64) and 28 (23.14%) had a very low (<55) PFI score.

Despite deviations from normality, the scatterplot of BMI versus PFI demonstrated a clear linear trend (Figure 3), confirmed by a strong negative Pearson correlation (r = −0.720, 95% CI = −0.796 to −0.622, *p* = 1.25 × 10^−20^) and consistent Spearman correlation (ρ = −0.659, *p* = 2.04 × 10^−16^). Bootstrap 95% CIs closely matched analytical estimates (Pearson = −0.803 to −0.612; Spearman = −0.765 to −0.517). In the adjusted linear regression model (PFI = β_0_ + β_1_·BMI + β_2_·sex), BMI remained a significant independent predictor (β = −3.13, 95% CI = −3.68 to −2.59, *p* < 0.001), while sex also contributed modestly (β = −2.69, 95% CI = −5.38 to −0.00, *p* = 0.050). VIF values (~1.00) confirmed the absence of multicollinearity, and residual diagnostics indicated no undue influence of single observations. A complementary categorical analysis showed a significant association between BMI group and PFI category (χ^2^(3) = 88.94, *p* < 0.001), with a large effect size (Cramér’s V = 0.86, 95% CI = 0.76 to 0.95). Stratified analyses by sex confirmed consistent inverse BMI–PFI relationships for both males (r = −0.69, *p* < 0.001) and females (r = −0.76, *p* < 0.001).

Overall, the results demonstrate a robust, inverse linear association between BMI and cardiorespiratory fitness as measured by the Harvard Step Test, consistent across sexes and supported by multiple complementary statistical approaches.

## 4. Discussion

Students’ physical fitness levels often decline due to the challenges of adapting to new social and academic dynamics, which can negatively impact their overall health [18]. This study aimed to evaluate the PFI of students at the State University of Novi Pazar using the modified HST and to examine its relationship with body composition, expressed through BMI. The research included 121 students with a mean BMI of 23.50, whose physical condition was assessed through the test. The results showed that 13.2% of students demonstrated good physical condition, 41.3% achieved high average scores, 19.0% showed low average scores, and 26.4% had poor physical condition. Statistical analysis revealed that students’ PFI was not satisfactory overall, given the high percentage of participants with low and poor results. This outcome may reflect the influence of sedentary lifestyles and insufficient participation in sports or structured physical activity. Furthermore, a significant difference in PFI was observed between students with a normal BMI and those classified as overweight, confirming the negative association between excess body weight and physical fitness.

A low level of physical fitness has been cited as a serious global problem in the last few decades. By analyzing about 50 studies, which included 25 million children aged 9 to 17 from 28 different countries in the period from 1964 to 2010, it was determined that the cardiovascular endurance performance of the current generation of children was significantly lower compared to previous generations. This decline can be explained by the presence of different aspects, including psychological, psychosocial, and physical [19].

A study conducted by Jabeen and Sarmila [20] gives us an insight into the unsatisfactory physical fitness results of a group of physiotherapy students. 50 students participated in that study, and as in our research, the modified HST was used to measure the PFI. It was also proven that subjects with a higher BMI had a lower level of PFI [20].

Based on the results of statistical analyses, our study reached a significant conclusion. Spearman’s correlation coefficient and the chi-square test were used to examine the relationship between BMI and PFI. A *p*-value of less than 0.001 indicates that it is statistically significant; that is, there is a negative correlation between BMI and PFI. The results of our study coincide with the results of numerous researchers, including Brunet and associates [21], who also emphasize how important it is to create early interventions that would improve physical fitness and stop the increase in obesity [21].

In a study by Arabmokhtari and associates [22] conducted among students aged 22 to 36, the relationship between BMI and cardiorespiratory fitness was considered. According to their findings, these two factors have a negative correlation, which means that cardiorespiratory fitness decreases with increasing BMI [22].

With strong evidence that higher levels of physical activity, especially when they exceed 150 min per week, result in greater weight loss, i.e., have a positive effect on reducing BMI, lack of activity leads to higher BMI values. Recent research confirms the inverse relationship between physical activity and BMI, emphasizing its role in weight management and inclusion as a key part of healthy lifestyle habits [23,24].

Similar research conducted among students and young adults showed that a higher percentage of body fat often correlates with poorer physical condition, because excess body weight leads to a decrease in endurance and functionality of the cardiorespiratory system. Based on the study conducted by Müller and associates [25], the aim of which was to examine the results of fitness and condition tests and their correlation with body composition parameters in 735 students aged 14 to 18 years, it was concluded that a higher percentage of adipose tissue led to lower PFI results.

The relationship between BMI, fat percentage, and VO_2_max was examined by Umamaheswari and associates [26] in persons aged 19 to 30 years with a BMI of >25 kg/m^2^. The results indicated that there is a positive correlation between fat-free mass and VO_2_max (*p* < 0.001) and an inverse correlation between VO_2_max and BMI (*p* < 0.001) [26].

The results of our study show that regular physical activity can significantly improve fitness levels and confirm the negative relationship between BMI and PFI. Higher BMI was associated with lower fitness levels, though causality cannot be inferred from the cross-sectional design. Future longitudinal or interventional studies are required to confirm causality and to explore mechanisms linking adiposity with cardiovascular fitness in young adults. However, these findings should be interpreted with caution, as the study was limited to students from a single university, employed a cross-sectional design, relied on BMI, which does not differentiate between fat and muscle mass [27], and assessed physical fitness using only one method [14].

Given the inverse correlation between BMI and physical fitness, and the increased prevalence of excess body mass (obesity) among students, it is recommended that their lifestyle be improved through structured programs of continuous physical activity. These may include aerobic exercise, resistance training, or flexibility exercises, implemented either individually or in group settings. In addition, student education on healthy and balanced nutrition is essential. This recommendation is supported by the findings of Plotnikoff and associates [28], who, in their meta-analysis, demonstrated that combined programs of physical activity and proper nutrition significantly improve physical fitness levels and encourage the adoption of healthier lifestyle habits among students.

Numerous studies further confirm the positive impact of aerobic activity on reducing BMI and improving overall health in young people. For example, Grace and associates [29] reported that a 10-week program of combined aerobic exercise resulted in a significant decrease in BMI, improvements in lipid profiles, and reductions in blood pressure. Similarly, Špirtović and associates [30], in their study of female students, found that aerobic training produced significant benefits in reducing BMI.

Overall, these approaches are effective in multiple ways. In addition to improving physical condition and reducing obesity through decreases in BMI, they also have motivational benefits for students. Most importantly, such interventions contribute not only to better health outcomes but also to improvements in students’ academic performance.

Our results extend current knowledge by providing context-specific evidence that can guide the development of university-based interventions in Serbia, including structured physical activity programs and wellness initiatives designed to improve students’ cardiovascular fitness and body composition.

### Study Limitations

This study has several limitations that should be acknowledged. First, the cross-sectional design prevents conclusions about causality between BMI and physical fitness, as only associations at a single time point were examined. Second, the sample was restricted to students from the University of Novi Pazar, which limits the generalizability of the findings to broader student or young adult populations. Third, BMI was used as a proxy for body composition and does not differentiate between fat mass, lean muscle, and bone tissue. Future studies employing direct measures such as bioimpedance or DXA would provide more precise insight into body composition–fitness relationships. Additionally, the exclusion of professional athletes, while intended to minimize bias, reduced the variability of the sample and may have limited the full range of fitness levels. The use of the modified HST as the only fitness measure also represents a limitation, as physical fitness is a multidimensional construct. Furthermore, performance on the test could have been influenced by participants’ familiarity with stepping rhythm and pulse counting, which may have introduced measurement error. Finally, other potentially important confounding variables, such as lifestyle habits, nutritional status, and socioeconomic background, were not controlled for, which may have influenced the observed results.

## 5. Conclusions

Our study revealed a negative relationship between PFI and BMI, implying that students with higher BMI have lower levels of PFI and vice versa. These associations should be interpreted with caution, as the cross-sectional design does not allow determination of cause and effect. Considering the existence of poor physical condition among students, this research emphasizes the need for regular exercise and physical activity, which is very important for the development and maintenance of physical condition, but also for the prevention of numerous diseases. As a result, it is crucial to motivate students to give priority to regular exercise and the development of good lifestyle habits that will not only affect their health, in general, but will also increase the level of their academic achievements. Given the relatively high prevalence of suboptimal physical fitness among university students, our findings serve as a critical wake-up call for public health authorities, emphasizing that urgent, targeted interventions are needed to reverse this trend and safeguard the health potential of the next generation.

## Figures and Tables

**Figure 1 jfmk-10-00449-f001:**
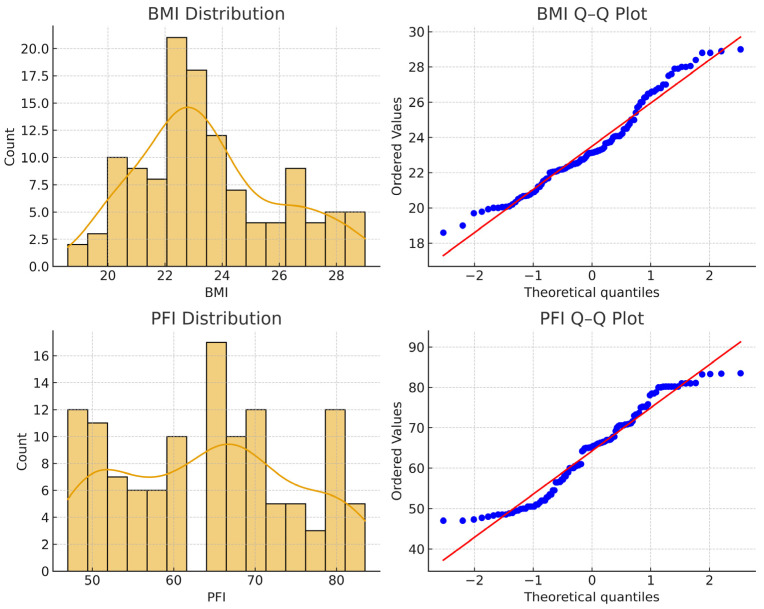
Normality assessment for Body Mass Index (BMI) and Physical Fitness Index (PFI): histograms with density curves and Q–Q plots (600 dpi). Both variables deviate slightly from perfect normality, particularly PFI.

**Figure 2 jfmk-10-00449-f002:**
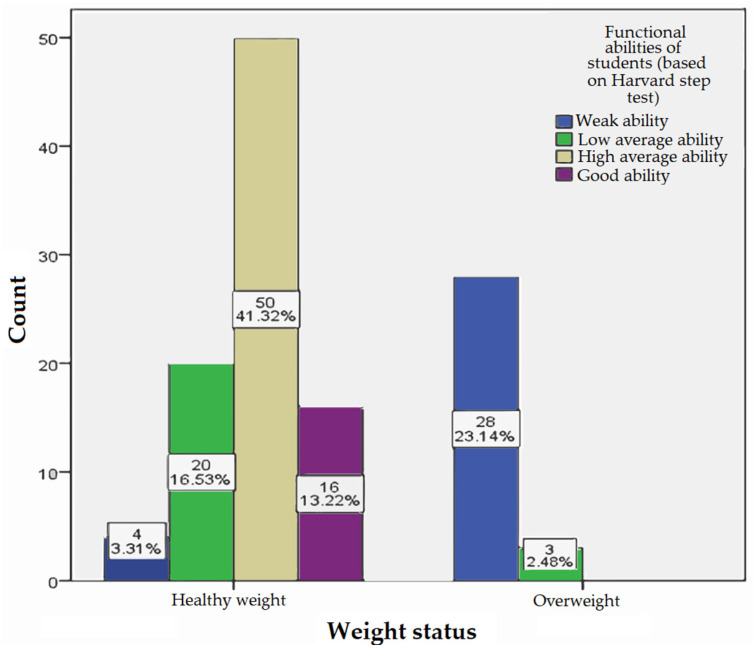
Frequency of Physical Fitness Index score.

**Figure 3 jfmk-10-00449-f003:**
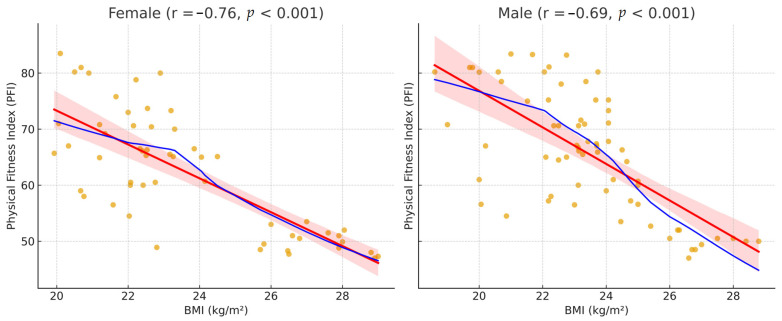
Scatterplots of BMI versus PFI stratified by sex, with linear (red) and LOESS (blue) fits (600 dpi). The orange dots represent individual BMI–PFI data points, while the pink shaded area shows the 95% confidence interval around the linear regression line. Both sexes show a consistent inverse relationship between BMI and Physical Fitness Index.

**Table 1 jfmk-10-00449-t001:** Descriptive statistics of research variables.

Variable	Mean	SD	Minimum	Maximum	Skewness	Kurtosis
Modified HST	64.233	10.784	47.00	84.00	0.057	−1.109
BMI	23.497	2.459	18.60	29.00	0.466	−0.457

Legend: SD—Standard Deviation; HST—Harvard Step Test; BMI—Body Mass Index.

**Table 2 jfmk-10-00449-t002:** Distribution of Physical Fitness Index and Body Mass Index.

Variable	Categories	N (%)	Mean ± SD
PFI	Good (80–89)	16 (13.2)	64.23 ± 10.78
	High average (65–79)	50 (41.3)	
	Low average (55–64)	23 (19.0)	
	Poor < 55	32 (26.4)	
	Total	121 (100)	
BMI	Normal weight (18.5–24.9)	90 (74.4)	23.50 ± 2.45
	Overweight (25–29.9)	31 (25.6)	

Legend: SD—Standard Deviation; PFI—Physical Fitness Index; BMI—Body Mass Index.

## Data Availability

The original contributions presented in this study are included in the article. Further inquiries can be directed to the corresponding author.

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
