# Peer review of "Relationship Between Physical Fitness Index and Body Mass Index: A Cross-Sectional Study in Serbian Students of Biomedical Sciences"

_jfmk, 2025, doi:10.3390/jfmk10040449_

Round 1
Reviewer 1 Report
Comments and Suggestions for Authors
The manuscript addresses a relevant public health topic and the relationship between physical fitness and BMI among university students is of continued interest. However, to meet the standards of a high-impact journal, several core aspects require clarification and strengthening, particularly regarding originality, methodological rigor, and interpretation of results.
Page 2, lines 55–64: The introduction provides general background but does not clearly identify the specific research gap. The negative association between BMI and physical fitness is well-documented; therefore, it is important to state what this study adds beyond confirming known findings in a new sample.
Recommendation: Add a final paragraph emphasizing the novelty.
Page 3, lines 104–112: The cross-sectional design is stated, but its limitations for drawing causal conclusions are not sufficiently acknowledged.
Recommendation: Explicitly mention that the results reflect associations only and do not allow directional inferences.
Page 3–4 (Modified Harvard Step Test): The Harvard Step Test is an accepted tool, but no justification is provided for its suitability over more current or multidimensional fitness assessments.
Recommendation: Briefly justify the use of this test based on its validity for this population.
Page 4, lines 144–152: No sample size or power calculation is provided. Although the study is exploratory, including a justification of sample adequacy would strengthen the methodological rigor.
Recommendation: Provide a short statement on expected effect size or reference similar studies.
Page 5, lines 165–172 (Table 1): The descriptive statistics are clear; however, confidence intervals are not reported. Including them would help contextualize the precision of estimates.
Page 6, lines 194–199: The negative correlation between BMI and fitness index is expected, but effect size interpretation (magnitude and practical significance) is not discussed.
Recommendation: Add a brief explanation on the strength and clinical relevance of the correlation.
Page 6–7, lines 204–218: The discussion restates the results but does not clearly articulate how these findings extend current knowledge.
Recommendation: Include a sentence explaining how the results may inform targeted interventions specific to university populations in Serbia.
Page 7, lines 283–291: The limitations section is useful but would benefit from explicitly mentioning the use of BMI rather than direct measures of body composition. This is critical as BMI does not differentiate fat mass from lean mass.
Recommendation: Include this point to enhance transparency regarding measurement limitations.
Page 8, lines 302–309: The conclusions are appropriate, but some language implies causality.
Recommendation: Rephrase to reflect association rather than causation.
The English language throughout the manuscript is generally understandable; however, several sections would benefit from editing to improve clarity, academic tone, and precision. In particular, some sentences are overly long or repetitive, and certain terms are used inconsistently. Improving sentence structure, eliminating redundancy, and enhancing the flow of ideas would greatly strengthen the overall readability and professionalism of the manuscript. I recommend a careful language revision, preferably by a native speaker or professional scientific editing service.
Author Response
Dear Reviewer,
Thank you very much for taking the time to review this manuscript and for your valuable contributions toward its substantial improvement.
Kind regards,
The authors

Reviewer 2 Report
Comments and Suggestions for Authors
The study addresses a clear and relevant question—the association between physical fitness (via a modified Harvard Step Test, HST) and body mass index (BMI) in university students. The topic is timely and of applied interest to health and kinesiology audiences. The manuscript is readable and generally well-structured (Introduction → Methods → Results → Discussion → Conclusions), with appropriate ethical approvals and informed consent reported. However, there are substantive methodological and analytic issues that currently limit the paper’s credibility and interpretability. In its present form, I would recommend major revisions prior to any consideration for publication.
The cross-sectional design is appropriate for describing associations but cannot support causal inferences; the Discussion and Conclusions occasionally verge on causal language or public-health exhortations that exceed what the design permits. More importantly, the operationalization of “physical fitness” as a single HST-derived index is too narrow for a construct as multidimensional as fitness. While the modified HST is a practical field test, the manuscript must (1) specify the exact step height used (and whether it differed by sex, as in the classical test), (2) confirm the cadence (steps/min) and test termination rules that were actually employed in this cohort, and (3) justify the PFI categorization thresholds with a primary source that matches the modified protocol used. At present, the Methods describe historical versions of the HST and an index formula but do not tie those precisely to the protocol executed in this sample; that gap undermines construct validity and comparability.
The sample comprises 121 students (18–25 years) from a single university and two study programs, excluding professional athletes. This is an understandable convenience sample, but several details are missing or imprecise: (a) the recruitment pathway (class announcements? email? flyers?) and participation rate, (b) the number screened and excluded (and for what reasons), and (c) missing-data handling (the text mentions excluding “incomplete or invalid questionnaire/test data,” but no counts or criteria for invalidity are provided). These omissions hinder reproducibility and may mask selection bias. Additionally, the text alternates between calling participants “adolescents” and “students/young adults”; please standardize terminology to reflect the actual age bracket (18–25 years) and avoid implying adolescent generalizability.
The HST heart-rate acquisition method (manual palpation vs. heart-rate monitor), the exact timing windows (first, second, third minute of recovery), and any inter-rater or intra-rater quality control are not reported. Given that the PFI is computed directly from those pulses, small counting/measurement errors can propagate into the index. If possible, report how pulse counts were obtained (device model if applicable), who measured them, and any practice trials. If no reliability study was conducted locally, cite and align with reliability data for the exact modified protocol used, and acknowledge the implications.
The BMI distribution contains only “normal weight” and “overweight” categories (no underweight or obesity classes). The text at one point describes the overweight group as “moderately obese,” which is incorrect; BMI 25.0–29.9 kg/m² is overweight, not obesity. This terminology must be corrected throughout. Because the BMI range is relatively narrow (mean ≈23.5, SD ≈2.45; min 18.6, max 29.0), any very high correlation with PFI warrants extra scrutiny for linearity, leverage points, and heteroscedasticity.
The statement that “there was no need to test for normality due to the sample size exceeding 30” is not statistically defensible. The central limit theorem pertains to sampling distributions of means, not to the distributional assumptions behind Pearson correlation (which assumes bivariate normality and linearity). You should: (1) report normality diagnostics (histograms/QQ plots and Shapiro-Wilk or D’Agostino tests) for both variables; (2) present a scatterplot of BMI vs. PFI with a linear fit and a nonparametric smoother (e.g., LOESS) to assess linearity; (3) provide Pearson’s r with 95% confidence intervals; and (4) complement this with Spearman’s ρ (already included) and a robust correlation (e.g., skipped correlation) if non-normality or outliers are suspected. Given the potential confounding by sex and study program (and possibly age within the 18–25 window), a multivariable linear regression is strongly advised (PFI = β0 + β1·BMI + β2·sex + β3·program + β4·age), with checks of residual diagnostics, multicollinearity, and influential observations (Cook’s distance, leverage, studentized residuals). Without these analyses, the reported association may partially reflect confounding structure rather than a BMI–fitness relationship per se.
The chi-square test reported (χ²=97.515; p reported as “0.000”) compares PFI categories across BMI groups. Two issues arise: (1) please format p as p < 0.001 and report degrees of freedom and expected counts to verify chi-square validity; and (2) dichotomizing or categorizing continuous variables discards information and can inflate apparent significance. If you retain the contingency analysis, report Cramér’s V with a confidence interval as the effect size and treat it as secondary to the continuous-variable correlation/regression.
The manuscript reports Pearson r = −0.720 and Spearman ρ = −0.659 between BMI and PFI—very strong effects for variables measured with modest precision in a healthy student sample. This may indeed reflect a real relationship, but the magnitude invites a transparency check: provide the scatterplot, influence diagnostics, and sex-stratified and program-stratified correlations (and possibly an interaction test BMI×sex) to ensure the effect is not driven by subgroup structure or a few high-leverage points. Also report confidence intervals for r and ρ.
No a priori sample-size calculation is provided. For transparency, you should add either (a) an a priori calculation based on a minimally important correlation (e.g., |r| = 0.25–0.30), two-sided α = 0.05, and 80–90% power; or (b) a post-hoc sensitivity analysis showing what effect sizes the study is actually powered to detect. As a helpful reference point: with N = 121 and α = 0.05 (two-sided), the study has ~80% power to detect correlations of about |r| ≈ 0.25; detecting |r| ≈ 0.30 would typically require ~84 participants, whereas detecting |r| ≈ 0.20 would require ~200. Stating this explicitly will calibrate the interpretation of your observed estimates and their precision.
Because the paper’s primary analysis is a single association, multiplicity is limited; however, the use of multiple parallel statistics (Pearson, Spearman, χ²), subgroup counts, and category tables benefits from a clear a priori primary outcome and primary analysis (e.g., Pearson correlation of continuous BMI and continuous PFI, confirmed by regression). Report all estimates with 95% CIs, not only p-values. Ensure all numerical claims in the Abstract match the Results, and avoid evaluative language such as “alarming prevalence” unless anchored in comparative data or prespecified thresholds.
Standardize the PFI category labels (e.g., “poor” vs. “very low”) so they align across Table 2, figure captions, and text. Ensure that category boundaries are exactly those of the cited source for the modified test and that you cite that source at first use. Make figure captions fully informative, and include the requested scatterplot. Prefer exact p-values (e.g., p = 0.003) where feasible, and avoid reporting “p = 0.000.”
Given known sex differences in both BMI and step-test performance, failing to adjust for sex (and potentially training status or study program) likely biases the crude correlation. The minimally acceptable model should include sex and program as covariates; sensitivity analyses could add age and interaction terms if justified. If adjustment materially reduces |r|, the Discussion should be tempered accordingly.
The Data Availability Statement promises raw data on request; for a straightforward cross-sectional dataset, consider supplying the anonymized dataset and analysis code as supplementary material. That will strengthen confidence in the unusually large observed correlation and facilitate reuse.
The manuscript tackles a pertinent question with a feasible field method in a relevant population. However, methodological and statistical reporting need substantial reinforcement before the findings are persuasive. Specifically, the paper requires: (1) precise and fully sourced HST protocol details (including step height, cadence, and PFI categorization aligned to the modified protocol actually used), (2) a clear analytic plan centered on a single primary analysis with appropriate assumption checks, visualization, effect sizes with CIs, and a multivariable regression adjusting for key confounders, (3) correction of BMI terminology and consistent category labels, (4) a sample-size justification or sensitivity analysis (with the note that N = 121 affords ~80% power for |r| ≈ 0.25), and (5) improved balance in the Discussion to reflect cross-sectional limits and the narrow BMI range in the sample. If the authors implement these changes and the association remains robust after adjustment and diagnostics, the work could be suitable for consideration as a brief original article.
Author Response

(The authors gave the same response as above.)

Round 2
Reviewer 2 Report
Comments and Suggestions for Authors
Dear Authors,
All my major and minor concerns have been corrected comprehensively and appropriately.
The revised manuscript demonstrates substantially improved methodological transparency, statistical rigor, and coherence.
I have no further questions.
Cheers.